# Screening, Synthesis and Biochemical Characterization of SARS-CoV-2 Protease Inhibitors

**DOI:** 10.3390/ijms241713491

**Published:** 2023-08-30

**Authors:** Martynas Bagdonas, Kamilė Čerepenkaitė, Aurelija Mickevičiūtė, Rūta Kananavičiūtė, Birutė Grybaitė, Kazimieras Anusevičius, Audronė Rukšėnaitė, Tautvydas Kojis, Marius Gedgaudas, Vytautas Mickevičius, Daumantas Matulis, Asta Zubrienė, Jurgita Matulienė

**Affiliations:** 1Department of Biothermodynamics and Drug Design, Institute of Biotechnology, Life Sciences Center, Vilnius University, Saulėtekio 7, LT-10257 Vilnius, Lithuania; martynas.bagdonas@gmc.vu.lt (M.B.); cerepenkaite.kamile@gmail.com (K.Č.); aurelija.mickeviciute@bti.vu.lt (A.M.); tautvydas.kojis@gmc.stud.vu.lt (T.K.); marius.gedgaudas@gmc.vu.lt (M.G.); daumantas.matulis@bti.vu.lt (D.M.); 2Department of Microbiology and Biotechnology, Institute of Biosciences, Life Sciences Center, Vilnius University, Saulėtekio 7, LT-10257 Vilnius, Lithuania; ruta.kananaviciute@gf.vu.lt; 3Department of Organic Chemistry, Kaunas University of Technology, Radvilenų pl. 19, LT-50254 Kaunas, Lithuania; birute.grybaite@ktu.lt (B.G.); kazimieras.anusevicius@ktu.lt (K.A.); vytautas.mickevicius@ktu.lt (V.M.); 4Department of Biological DNA Modification, Institute of Biotechnology, Life Sciences Center, Vilnius University, LT-10257 Vilnius, Lithuania; audrone.ruksenaite@bti.vu.lt

**Keywords:** SARS-CoV-2, papain-like protease, main protease, thiazole, disulfide, thermal shift assay, differential scanning fluorimetry, inhibition

## Abstract

The severe acute respiratory syndrome-causing coronavirus 2 (SARS-CoV-2) papain-like protease (PL^pro^) and main protease (M^pro^) play an important role in viral replication events and are important targets for anti-coronavirus drug discovery. In search of these protease inhibitors, we screened a library of 1300 compounds using a fluorescence thermal shift assay (FTSA) and identified 53 hits that thermally stabilized or destabilized PL^pro^. The hit compounds structurally belonged to two classes of small molecules: thiazole derivatives and symmetrical disulfide compounds. Compound dissociation constants (K_d_) were determined using an enzymatic inhibition method. Seven aromatic disulfide compounds were identified as efficient PL^pro^ inhibitors with K_d_ values in the micromolar range. Two disulfides displayed six-fold higher potency for PL^pro^ (K_d_ = 0.5 µM) than for M^pro^. The disulfide derivatives bound covalently to both proteases, as confirmed through mass spectrometry. The identified compounds can serve as lead compounds for further chemical optimization toward anti-COVID-19 drugs.

## 1. Introduction

SARS-CoV-2 detected in China in December 2019 spread rapidly worldwide and was accompanied by high infectivity and mortality rates [1]. The COVID-19 pandemic has revealed that despite advances in the medical biotechnology and pharmaceutical industry sectors, there is still a great need to study coronavirus mutation, spread and proteins to understand their action and to develop an effective treatment to stop future epidemic outbreaks [2,3,4].

Since the coronavirus infection outbreak, several vaccines have been approved, and vaccination has emerged as one of the most effective ways to prevent the spread of the coronavirus [5]. However, spike protein mutations led to the emergence of new strains of the coronavirus, against which vaccines were less effective. Another strategy for treating the acute respiratory disease caused by coronavirus is the development of antiviral drugs, which could suppress virus multiplication in the human body and thus alleviate the symptoms of the disease. Numerous drug discovery projects initiated by biopharmaceutical companies and public sector organizations have emerged to find promising antiviral lead compounds, and more than 700 agents with anti-SARS-CoV-2 activities reported in preclinical or clinical studies were reviewed in [6,7,8]. All of the viral proteins involved in its life cycle were possible targets for anti-coronavirus drug development. However, papain-like protease (PL^pro^) and main protease (M^pro^, also known as 3CL^pro^) have emerged as the most important targets to inhibit viral activity.

Both PL^pro^ and M^pro^ are cysteine hydrolases capable of cleaving polyproteins pp1a and pp1b at multiple sites to release sixteen non-structural proteins (nsp). PL^pro^ cleaves sites between nsp1 and 2, nsp2 and 3, and nsp3 and 4, while M^pro^ cuts the other 11 sites between the remaining nsps, resulting in a total of 16 nsps [9]. These nsps assemble and form the viral replication and transcription complex on the host cell membrane. During SARS-CoV-2 infections, such cleavage of proteins led to enhanced cytokine production and the inflammatory response observed in COVID-19 patients [10,11].

The active site of SARS-CoV-2 PL^pro^ features a catalytic triad formed by Cys111, His272, and Asp286 that cleaves the LXGG↓XX motif in viral proteins (nsp1–nsp4). Cys111 acts as the nucleophile, His272 as a general acid-base, and Asp286 favors the His alignment, thus promoting Cys111 deprotonation [12]. In addition, PL^pro^ can act as deubiquitinase. The enzyme cleaves the C-terminal LXGG sequence of ubiquitin and ubiquitin-like interferon-stimulated gene 15 (ISG15) protein leading to the suppression of the innate immune response. Thus, targeting PL^pro^ can inhibit viral replication and promote antiviral immunity [13].

The active site of M^pro^ consists of a catalytic dyad comprised of a nucleophilic cysteine Cys145 and nearby histidine His41 [14]. M^pro^ cleavage motif Gln↓(Ser/Ala/Asn) is highly specific for this enzyme, although some human cysteine proteases (cathepsins B, K, L, S) can also cleave after glutamine [15]. The dimerization of M^pro^ is essential for catalytic activity as the N-finger of each protomer interacts with the Glu166 of the other protomer, forming the S1 pocket of the substrate-binding site [16].

Several of the most recent reviews provide an overview of the SARS-CoV-2 protease inhibitors that have been approved or are being investigated for infection treatment, highlighting their chemical structures and binding modes [17,18,19]. The known PL^pro^ inhibitors were mainly developed based on GRL0617. This naphthalene scaffold-containing compound was originally developed as a non-covalent inhibitor of SARS-CoV PL^pro^, and in 2020, was found to inhibit the SARS-CoV-2 PL^pro^ protease. The efficiency of this compound in inhibiting PL^pro^ activity is not high (IC_50_ about 2 μM) [13,20]. The surface plasmon resonance (SPR) results showed weak GRL0617 binding to PL^pro^ (*K*_d_ 10.8 μM). Several new compounds with naphthyl subunits were synthesized, showing more potent binding to PL^pro^ (*K*_d_ of 2.6 μM) [21]. The restricted binding pockets at substrate binding sites (Gly-Gly recognition) were a key reason for the lack of potent PL^pro^ inhibitors. Due to their insufficient efficacy, none of the PL^pro^ inhibitors have entered the clinical trial stage. Recently, several covalent inhibitors were discovered with a modified methyl group of GRL0617 by an electrophile capable of reacting with Cys111 of PL^pro^. These covalent inhibitors seem to be the most efficient inhibitors of PL^pro^ discovered to date, with *IC*_50_ exceeding 0.1 μM [22].

A significantly larger number of designed compounds with inhibitory properties toward M^pro^ than PL^pro^ have been reported, including covalent peptidomimetic and non-peptidomimetic and non-covalent inhibitors targeting substrate binding and allosteric sites [6,23,24,25,26]. The most successful antiviral agent, Paxlovid, is the only FDA-approved drug for the treatment of high-risk patients with a confirmed SARS-CoV-2 infection. One of Paxlovid’s components is nirmatrelvir, a covalent peptidomimetic compound in which a nitrile warhead forms a covalent bond with Cys145 of M^pro^. Nirmatrelvir demonstrates high affinity (*IC*_50_ = 4 nM) and selectivity for M^pro^ versus human cysteine proteases [27]. Another non-covalent and non-peptidomimetic compound, S-217622, was found to be a nanomolar inhibitor of M^pro^ (*K*_d_ 13 nM) [28] and is currently in a phase 2/3 study in Japan. The treatment of patients with mild-to-moderate COVID-19 with S-217622 (ensitrelvir) demonstrated rapid and favorable antiviral efficacy and an acceptable safety profile [29]. The efficacy of ensitrelvir in a wide range of patients, including older people, with COVID-19 will be further assessed in a phase 3 multinational study.

Many compounds were in silico screening as potential dual inhibitors of SARS-CoV-2 proteases. However, the dual inhibitory effect was only proven for some compounds through in vitro experiments. One compound with reactive α-chloroketone moiety was identified as a dual-acting protease inhibitor with a micromolar affinity for both proteases (*IC*_50_ for M^pro^ 1.72 µM, *IC*_50_ for PL^pro^ 0.67 µM) [30]. In the study by Meewan et al. [31], several thiuram disulfide or dithiobis-(thioformate) derivatives inhibited three proteases—M^pro^, PL^pro^, and human cathepsin L—in the sub-micromolar range.

In this work, we screened a library of 1300 compounds against SARS-CoV-2 PL^pro^ protease and tested several of the most potent hit compounds’ efficacy in inhibiting PL^pro^ enzymatic activity. The two resultant groups, the thiazole- and disulfide-containing compounds that inhibit PL^pro^, were tested as M^pro^ inhibitors to probe the possibility of dual action by these compounds.

## 2. Results

A library of the available low molecular weight (MW) compounds was screened using a fluorescent thermal shift assay (FTSA) to detect their binding to a full-length recombinant protein PL^pro^ by monitoring its melting temperature (*T_m_*) change upon compound addition. To accurately measure the PL^pro^ melting temperatures, we used several dyes, including ANS and Glomelt, which increase the fluorescence upon the appearance of exposed hydrophobic protein amino acids when the protein unfolds. We estimated the lowest suitable protein concentration (5 µM) and the final dye concentration (50 µM of ANS) to be optimal for compound screening. All of the PL^pro^ melting curves demonstrated a single melting transition. Some hit compounds exhibited thermal stabilization of the protein, thus indicating compound binding.

Compounds were selected for screening from 1300 small molecules available from our in-house library. Most of the compounds were primary or secondary sulfonamides that have been synthesized in our laboratory or in collaboration with other laboratories and designed as carbonic anhydrase inhibitors. Furthermore, many resorcinol-bearing, heterocyclic compounds recently investigated as possible Hsp90, HDAC, or MMPs inhibitors were included in this library. The structures of over 550 compounds from this library and their interactions with carbonic anhydrases and Hsp90 were provided in the Protein-Ligand Binding Database (PLBD), available at https://plbd.org (accessed on 15 July 2023).

The compounds were screened at concentrations of 200 μM and 50 μM. Compounds that stabilized PL^pro^ and induced a melting temperature shift of >1.5 °C or had a large destabilizing effect were selected for follow-up studies. Compounds that produced poorly defined melting curves were excluded from further analysis. More than 53 positives, called “hits”, were identified based on this threshold value.

We tested all of the confirmed hits in dose-response biochemical inhibition experiments. The affinity (*K*_d_) of the hit compounds for PL^pro^ was evaluated using Z-Arg-Leu-Arg-Gly-Gly-AMC acetate as a substrate. The fluorescence of the released product, namely C-terminal AMC dye (7-amido-4-methylcoumarin), generated by PL^pro^ proteolysis was monitored at an excitation wavelength of 340 nm and an emission wavelength of 450 nm. Compounds that did not show inhibition at the highest used concentration of 200 µM were defined as inactive (*K_d_* ≥ 200 μM). Hits that inhibited less than 50% of the enzymatic activity at 50 µM of the compound were not further investigated in the compound dose-response assay due to weak potency. However, we determined the exact *K*_d_ values for several such compounds to elucidate the structure–activity relationship (SAR) of the inhibitors and to identify the functional groups that govern enhanced affinity. Structurally similar hits were grouped into compounds containing a naphthalene ring (Figure 1A) and ligands containing a disulfide bond (Figure 1B). Structural similarities between the compounds in each group guided the SAR studies.

### 2.1. Naphthalene-Based PL^pro^ and M^pro^ Inhibitors

Based on the modifications of 3-[naphthalene-1-yl(1,3-thiazol-2-yl)amino]propanoic acid (compound **3**), the compounds were grouped into compounds bearing (Figure 2):(i)substituents at the 4- position on the thiazole ring (compounds **4**–**12**);(ii)substituents at the 4,5 positions on the thiazole ring (compounds **13**–**24**);(iii)replacing the naphthalene group of compound **3** with the 4-methyl- or 4-aminophenyl group and introducing different substituents at the 4,5 positions of the thiazole ring (compounds **25**–**31**).

**Figure 2 ijms-24-13491-f002:**
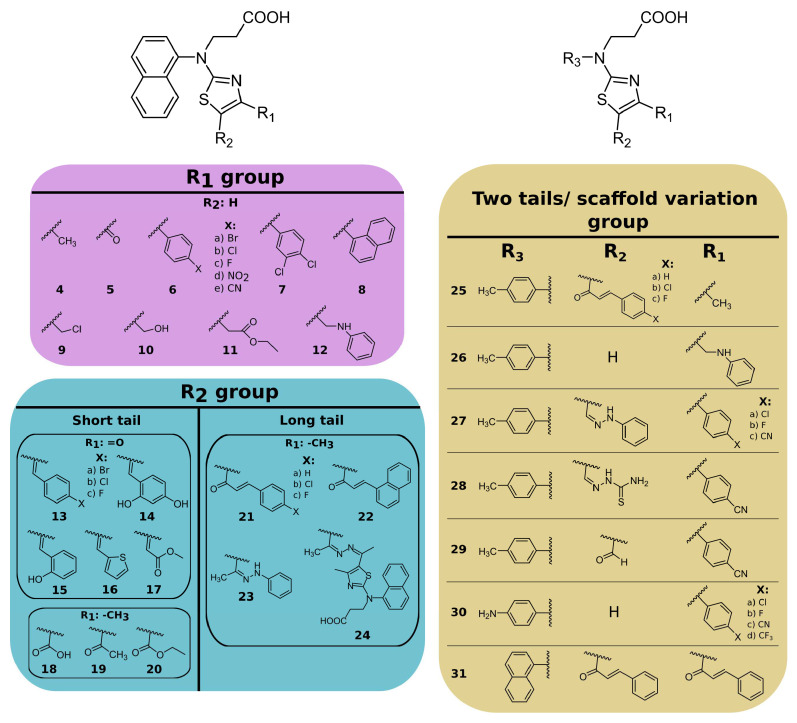
Chemical structures of thiazole derivatives. Compound synthesis is described in the Appendix A.

Nineteen compounds inhibited PL^pro^ activity with a *K*_d_ value of ≤50 μM (Table 1, Appendix A). We performed a detailed structure–activity correlation analysis for this group of compounds and found that the parent compound, 3-(naphtylamino]propanoic acid (compound **1**), weakly inhibited the PL^pro^ activity (*K*_d_ = 68 µM). The attachment of the thioamide functional group to the nitrogen atom (compound **2**) led to a 2-fold increase in the binding affinity (*K*_d_ = 29 µM), while the introduction of a thiazole ring (compound **3**) enhanced the affinity by 5-fold (*K_d_* = 14 µM) compared to the starting compound **1** (Figure 3).

Examining the binding of the first group of compounds modified in the 4-position of the thiazole ring, it is evident that all of the modifications decreased the binding affinity. For example, the addition of the methyl group (compound **4**) weakened the interaction by 8-fold, while carbonyl (compound **5**) resulted in an inactive compound (*K*_d_ > 200 µM). Only one compound, **6e**, with a 4-cyanophenyl substituent at the 4-position, bound PL^pro^ with similar affinity to the parent compound **3** (*K*_d_ 20 µM and 14 µM, respectively).

Compounds with 4,5-substituents on the thiazole ring can be divided into two subgroups: ligands with relatively short (**13**–**20**) or long (**21–24**) tails at the 5-position (R2 substituents). These compounds have carbonyl or methyl groups (R1 substituent) at the 4-position. In the subset of compounds with a short R2 substituent (**13**–**20**), most ligands bound to PL^pro^ weakly (*K*_d_ ≥ 50). However, compounds **13c** (Figure 4) and **14,** with fluorinated or dihydroxylated benzylidene substituents, interacted with higher affinity (*K*_d_ 8.6 µM and 9.2 µM, respectively). An interesting trend was observed among the long-tail–bearing compounds (**21**–**24**). The attachment of the styryl ketone functional group did not affect the binding (*K*_d_ (**21a**) = 110 µM). However, the introduction of a fluorine atom into the para position of styryl ketone (**21c**) strengthened the interaction by two times (*K*_d_ (**21c**) = 52 µM). Fluorine substitution by chlorine at the same position led to a 4-fold improvement in the inhibition (*K*_d_ (**21b**) = 29 µM). Interestingly, the styryl ketone group modification with (naphthalen-1-yl)acryloyl fragment (**21a**–**22**) resulted in the most potent inhibitory activity against PL^pro^ (*K*_d_ = 5.2 µM) in this series of compounds (Figure 5).

The third group of compounds (**25**–**31**) bearing 4-methyl- or 4-aminophenyl groups (as R3 substituent) instead of naphthalene were weak PL^pro^ inhibitors. The affinity for compound **28** with a modified thiosemicarbazide substituent at the R2 position and 4-methylphenyl group at the R3 position was the highest (*K*_d_ = 8.8 µM) from this group of compounds. In most cases, naphthalene modification by 4-methylphenyl groups only had a negligible effect on the binding affinity. For example, the affinity of naphthalene-containing compound **21b** was 2-fold higher than analogous 4-methylphenyl-containing compound **25b** (*K*_d_ 29 µM and 66 µM, respectively). Conversely, naphthalene-bearing **12** modification with the 4-methylphenyl group (compound **26**) showed a 2-fold improvement (*K*_d_ 29 µM and 16 µM, respectively). Also, the modification of the naphthalene ring by the 4-aminophenyl group did not improve the binding affinity (*K*_d_ for **30c** was 32 μM, while *K*_d_ for **6c** was 20 μM). These results showed that variations in the main structural element had no significant effect on the binding.

We next determined whether our PL^pro^ inhibitors inhibited the M^pro^ activity using ([5-FAM]-AVLQSGFR-[Lys(Dabcyl)]-K-amide as a substrate. Interestingly, six thiazole derivatives were inhibitors of M^pro^, with **13a** and **13c** being the most potent (*K*_d_ 6.5 µM and 8.6 µM, respectively) (Table 1). The *K_d_* values indicated that **13a** was 3.7-fold more potent for M^pro^ inhibition than PL^pro^, and **13c** bound both proteases with the same affinity (*K*_d_s are 8.6 µM, Figure 4).

Overall, the SAR results emphasized the thiazole derivatives as a promising scaffold for the further development of dual PL^pro^ and M^pro^ inhibitors.

### 2.2. Disulfide Derivatives as PL^pro^ and M^pro^ Inhibitors

We found seven benzene disulfide compounds with varied functional groups (Figure 6) that were efficient PL^pro^ inhibitors, with *K*_d_ values between 0.43 and 2.5 μM (Table 2). The highest affinities (*K*_d_ near 0.5 μM) were determined for compounds **32**, **34,** and **37**. Moreover, two compounds—**32** and **34**—inhibited M^pro^ with *K*_d_ values of 2.8 and 3.5 μM, respectively. Compound **34** was eight times more potent for PL^pro^ than M^pro^ (Figure 7).

It was recently confirmed through native mass spectrometry that disulfiram and tetraethylthiuram disulfide interact covalently with cysteine residues of M^pro^ and the inhibition potency highly depends on the reducing agents, such as DTT or glutathione [33]. However, Wang et al. [34] synthesized unsymmetrical aromatic disulfide derivatives, inhibiting SARS-CoV M^pro^ through the reversible binding mechanism. We performed mass spectrometry experiments to elucidate the covalent binding mechanism of disulfide compounds with the proteases in the absence of reducing agents. The covalent binding of **32** with PL^pro^ and M^pro^ was confirmed through MS experiments and the data are shown in Figure 8. Compound **32** (MW (**32**) = 248.37) incubated with PL^pro^ showed a major peak relating to protein–compound adduct, with a mass shift equal to the compound molecular mass of half moiety of the disulfide (ΔMW = 38,507.79 − 38,384.66 = 123.13) and a minor set of peaks relating to two or three inhibitor molecules bound to the protein. The MS data also showed the covalent binding of compound **34** to PLpro (Appendix A). The molecular weight of PL^pro^ changes upon the addition of ligands **32** and **34** by approximately 8.6 Da (from 38,393.27 to 38,384.66), it is most likely that with this addition, modification occurs in the presence of disulfide compounds.

The molecular weight of M^pro^ after incubation with compound **32** had a major shift corresponding to two bound inhibitor molecules with their half moieties (ΔMW = 34,040.75 − 33,795.02 = 245.7). Also, minor peaks relating to one and three inhibitor molecules bound to the protein were detected. The MS results confirmed that symmetrical disulfide compounds can modify the catalytic cysteine (C111 of PL^pro^ and C145 of M^pro^) and bind to other cysteine residues in the proteases.

To confirm the covalent nature of the disulfide compound binding to PL^pro^, we produced mutant PL^pro^ C111S and performed FTSA experiments with disulfide **32** and GRL0617. The binding affinities of wt PL^pro^ and the mutant PL^pro^C111S proteins for GRL0617, a well-known noncovalent inhibitor, were similar (*K*_d_ 1.2 and 2.5 μM, respectively). The addition of the disulfide compound **32** strongly destabilized the native PL^pro^, but not the mutant PL^pro^C111S (Figure 9A,B). This shows that the binding of disulfide **32** involves catalytic cysteine and the covalent modification of cysteine would lead to protein degradation.

## 3. Discussion

In the search for compounds that would exhibit inhibitory properties against any enzymes participating in any disease, a library of compounds is often screened to discover whether any of the compounds bind with sufficient affinity. Various assays are used for screening and may be based on recombinantly prepared target proteins or cells containing the proteins. Both inhibition and binding assays may be employed. Here, we chose the fluorescence-based thermal shift assay (FTSA, also known as differential scanning fluorimetry, DSF) as it requires low protein consumption and simple PCR equipment that allows for fluorescence measurements over increased temperatures. Even weakly binding ligands affect protein stability, and based on a shift in the melting temperature (*T_m_*) of the target protein, hit compounds can be identified. Additional molecular interactions between ligand and protein usually stabilizes the protein, thus reducing the Gibbs energy [36,37]. Compounds that bind more strongly will exhibit greater thermal stabilization of the proteins than weakly binding compounds.

However, in addition to thermal stabilization, the same target protein may exhibit thermal destabilization in the presence of some ligands, shifting the *T_m_* downward. It is thought that such ligands bind more strongly to the unfolded than to the native protein state. However, the destabilizing effect is still poorly understood [38]. Compounds that destabilize a protein may lead to more rapid protein degradation. Therefore, such destabilizers are also worth investigating [39]. Our studies confirmed that protein destabilizers, such as symmetrical disulfide compounds, could be potent SARS-CoV-2 PL^pro^ and M^pro^ inhibitors. However, their use as SARS-CoV-2 drugs could be limited due to the reactive nature of such compounds. The disulfide compounds not only modify amino acids at the active site, but can most likely bind to nucleophylic amino acids localized in other sites of the protein.

The structure–activity relationship of the naphthalene-bearing thiazole derivatives revealed that the naphthalene group is not a necessary structural element for PL^pro^ inhibition. It was previously shown through the X-ray crystallography of GRL0617 and its derivatives in complex with PL^pro^ that the naphthyl ring occupies the hydrophobic pocket of the enzyme and forms hydrophobic interactions with the aromatic amino acids Y264 and Y268 [40]. The replacement of naphthyl by the benzylidene group had similar effects to the binding affinity. Compounds such as **13c** with benzylidene moiety could inhibit PL^pro^ with *K*_d_ 8.6 µM, while the best naphthalene-group bearing compound, **22,** inhibited PL^pro^ with *K*_d_ 5.2 µM. It should be emphasized that the binding affinity to both proteases was highly dependent on the nature, length, and hydrophobicity of the substituents on the thiazole ring. The identified hit compounds were quite varied in their structures; therefore, it was difficult to perform detailed SAR analysis. We believe thiazole-bearing compounds can be optimized further by modifying the carboxy-ethyl group. In our library of compounds, there were no modifications at this site, but further optimization at the 2‘-position on the thiazole ring might improve the affinity of compounds for both SARS-CoV-2 proteases.

## 4. Materials and Methods

### 4.1. Chemistry

Commercially available solvents and reagents were used without further purification unless otherwise mentioned. Reagents and solvents were obtained from Sigma-Aldrich (St. Louis, MO, USA) and used without further purification. The reaction course and purity of the synthesized compounds were monitored through TLC using aluminium plates precoated with silica gel with F_254_ nm (Merck KGaA, Darmstadt, Germany). Melting points were determined with a B-540 melting point analyzer (BüchiCorporation, New Castle, DE, USA) and were uncorrected. NMR spectra were recorded on a Brucker Avance III (400, 101 MHz) spectrometer. Chemical shifts were reported in (δ) ppm relative to tetramethyl silane (TMS) with the residual solvent as an internal reference (DMSO-*d_6_*, δ = 2.50 ppm for ^1^H and δ = 39.5 ppm for ^13^C). Data were reported as follows: chemical shift, multiplicity, coupling constant (Hz), integration, and assignment. IR spectra (ν, cm^−1^) were recorded on a Perkin–Elmer Spectrum BX FT–IR spectrometer (Perkin–Elmer Inc., Waltham, MA, USA) using KBr pellets. Elemental analyses (C, H, N) were conducted using the Elemental Analyzer CE-440 (Exeter Analytical, Inc., Chelmsford, MA, USA); their results were found to be in good agreement (±0.3%) with the calculated values.

Synthetic details for the preparation of compounds **1**–**38** and the characterization data of these compounds are given in the Appendix A.

### 4.2. Protein Expression and Purification

The initial *Escherichia coli* codon-optimized plasmid pGBW-m4046920 was a gift from Ginkgo Bioworks and Benjie Chen (Addgene plasmid # 145717; http://n2t.net/addgene:145717 (accessed on 15 July 2023); RRID: Adgene_145717). Using pET28b(+) vector (Novagen), we created a plasmid encoding full-length PL^pro^ (747–1065 a.a. fragment of the nsp3) with N-terminal 6xHis-tag, separated from protein by a thrombin cleavage site. The plasmid was expressed in *E. coli* BL21(DE3) strain (Novagen). Cells were grown in LB medium until OD600 ~ 0.7. Expression was induced using 0.25 mM IPTG. At the same time, 1 mM ZnCl_2_ was added. Cells were then incubated overnight at 22 °C and harvested through centrifugation. SDS-PAGE was performed to confirm PL^pro^ expression. Protein was purified using Ni-IDA-sepharose. Gradient elution was created using buffer A (25 mM TRIS, 0.5 M NaCl, 70 mM imidazole, 5% glycerol, 2 mM β-ME, pH 7.5) and buffer B (25 mM TRIS, 0.5 M NaCl, 0.5 M imidazole, 5% glycerol, 2 mM β-ME, pH 7.5). SDS-PAGE was performed to confirm the purity of the elution fractions and the protein was dialyzed overnight in dialysis buffer (20 mM HEPES, 50 mM NaCl, 2 mM DTT, 5% glycerol, 1 mM EDTA, pH 7.5). Then, the buffer was changed into storage buffer (20 mM HEPES, 50 mM NaCl, 2 mM DTT, 5% glycerol, pH 7.5) and dialyzed for two more hours. Protein was stored at −80 °C.

Site-directed mutagenesis of plasmid encoding full-length PL^pro^ was performed with primers (Appendix A) using the standard PCR procedure and verified through sequencing analysis (Appendix A). Expression and purification conditions for the PL^pro^C111S mutant were the same as for the native protein.

Recombinant full-length M^pro^ protein was purchased from Sigma-Aldrich.

### 4.3. Fluorescence Thermal Shift Assay (FTSA)

FTSA experiments were performed with the real-time PCR instrument “QIAGEN Rotor-Gene Q”. The PL^pro^ solution in the absence and presence of compounds was heated from 25 °C to 99 °C through increasing the temperature by 1 °C per minute. We optimized the experimental conditions for the screening of compounds: assay buffer, PL^pro^ concentration, and fluorescence dye (1-anilino-8-naphthalenesulfonate (ANS) or Glomelt). The fluorescence excitation wavelengths for ANS and Glomelt were set at 365 or 468 nm and the detection wavelengths were 460 or 507 nm, respectively. The optimal PL^pro^ concentration was 5 µM and ANS concentration 50 µM. In addition, 10 mM stock solutions of 1300 compounds from our in-house library were made in DMSO, then diluted in 50 mM HEPES containing 100 mM NaCl at pH 7.5, with the final compound concentration of 50 and 200 μM. Protein solutions were prepared in 5 mM HEPES (pH 7.5) containing 10 mM NaCl. Protein *T_m_* values were obtained by fitting the melting curves to a two-state model.

To evaluate the binding affinity, 2-fold serial dilutions of the ligand stock in DMSO were performed, and then the prepared samples were 50-fold diluted with the buffer. Data analysis and compound dissociation constants were determined as previously described [35].

### 4.4. Enzymatic Inhibition Assay

The protease enzymatic activity of SARS-CoV-2 PL^pro^ was measured using a FRET-based enzymatic activity assay. Experiments were carried out in non-binding 96-well plates at 25 °C, containing a final reaction volume of 100 µL. Ligand solutions (200–0 µM) were prepared in 50 mM HEPES buffer containing 100 mM NaCl, 2% (*v*/*v*) DMSO, pH 7.5, and mixed with PL^pro^ (final concentration 40 nM). After 30 min of incubation, peptidomimetic substrate Z RLRGG-AMC (250 µM) was added to initiate the reaction. Fluorescence was monitored for 30 min on a BioTek Synergy H4 Hybrid plate reader (λex = 340 nm; λem = 450 nm). Enzyme activity was equalized to the slope of the reaction progress linear curve and relative activity was calculated from the slopes’ ratio of inhibited and control (without ligand) samples. Three independent experiments with each ligand (except for low potency compounds, *K*_d_ > 50 µM) were used to calculate mean relative enzyme activity. Mean relative activity values were applied for the evaluation of binding affinities using the Morrison Equation (1). Data precision was evaluated using standard deviation.
(1)$$\mathrm{Enzyme}\;\mathrm{activity}\;\left(\%\right)=\left(1-\frac{\left(\left[E\right]+\left[I\right]+K_{d}\right)-\sqrt{{\left(\left[E\right]+\left[I\right]+K_{d}\right)}^{2}-4\cdot \left[E\right]\cdot \left[I\right]}}{2\cdot \left[E\right]}\right)\cdot 100$$

Inhibition experiments of compounds **32**–**38** were performed with PL^pro^ dialyzed into 20 mM HEPES buffer containing 50 mM NaCl, 5% glycerol, pH 7.5 to avoid DTT impact for disulfide compounds stability disruption.

The same procedure was used for M^pro^ enzymatic activity determination. Ligand samples were prepared in 20 mM HEPES buffer containing 50 mM NaCl, 10% glycerol, 2% (*v*/*v*) DMSO, pH 7.3, and mixed with M^pro^ (final concentration 50 nM). After 30 min of incubation, fluorogenic substrate [5-FAM]-AVLQSGFR-[Lys(Dabcyl)]-K-amide (3 µM) was added to initiate the reaction. Fluorescence was monitored for 40 min (λex = 480 nm; λem = 520 nm).

### 4.5. Mass Spectrometry

Mass spectrometry experiments were performed with an electrospray ionization time-of-flight mass spectrometer (Q-TOF) to show the covalent binding of compound **32** to proteases. The 0.1 mg/mL protease solution was prepared in the absence or presence of **32** (1:5 PL^pro^: compound molar ratio and 1:10 M^pro^: compound molar ratio) and the solution was incubated for 1 h at room temperature before analysis. The final DMSO concentration was 1% (*v*/*v*).

## 5. Conclusions

A new class of inhibitors of SARS-causing coronavirus CoV-2 protease PL^pro^ were discovered using the thermal shift assay. From a library of 1300 available compounds, 53 hits were selected for the in vitro enzyme inhibition assay of PL^pro^. Several thiazole-bearing compounds inhibited PL^pro^ with micromolar affinity. Further optimization at the 2′-position on the thiazole ring is required to improve the activity of the compounds as SARS-CoV-2 proteases inhibitors. Several symmetrical disulfide compounds with a dissociation constant in the sub-micromolar range (*K*_d_ = 0.5 µM) were identified. Interestingly, two disulfides also demonstrated promising M^pro^ inhibition, with *K_d_* values of 3.5 µM. The mass spectrometry data revealed the covalent binding mechanism of disulfides toward both proteases. The identified compounds with thiazole or aromatic disulfide scaffolds can serve as lead compounds for anti-coronavirus drug development.

## Figures and Tables

**Figure 1 ijms-24-13491-f001:**
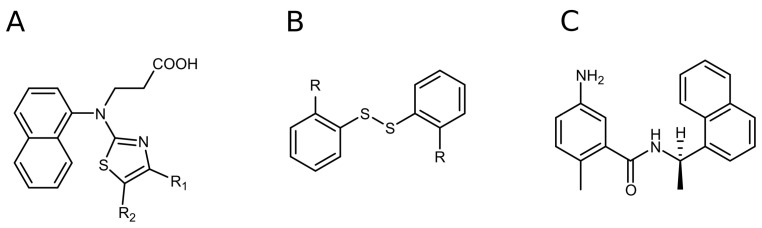
The discovered PL^pro^ hits: naphthalene-based (**A**) and disulfide bond-bearing compounds (**B**). Structure of GRL0617, one of the first compounds discovered as a selective SARS-CoV-2 PL^pro^ inhibitor (**C**).

**Figure 3 ijms-24-13491-f003:**
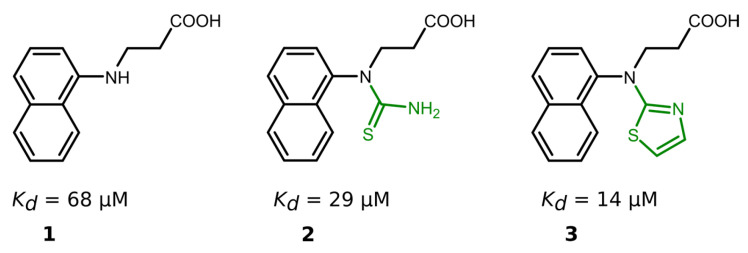
Compound binding affinity for PL^pro^. The addition of thiazole group (**1**–**3**) increased the binding affinity for PL^pro^ by approximately 5-fold.

**Figure 4 ijms-24-13491-f004:**
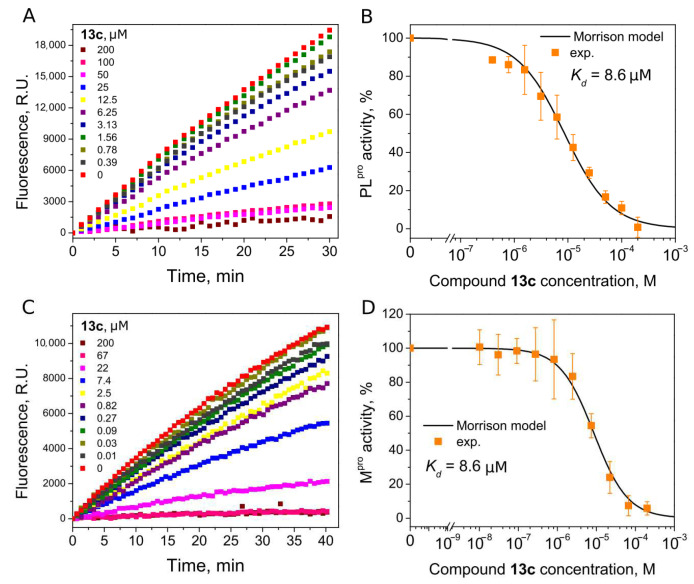
Inhibition of PL^pro^ (**A**,**B**) and M^pro^ (**C**,**D**) activity by compound **13c**. (**A**,**C**) Time-dependent product formation in the absence and presence of inhibitor. (**B**,**D**) Dose–response curve of PL^pro^ and M^pro^ inhibition with **13c** obtained after fitting curves in panels (**A**,**C**).

**Figure 5 ijms-24-13491-f005:**
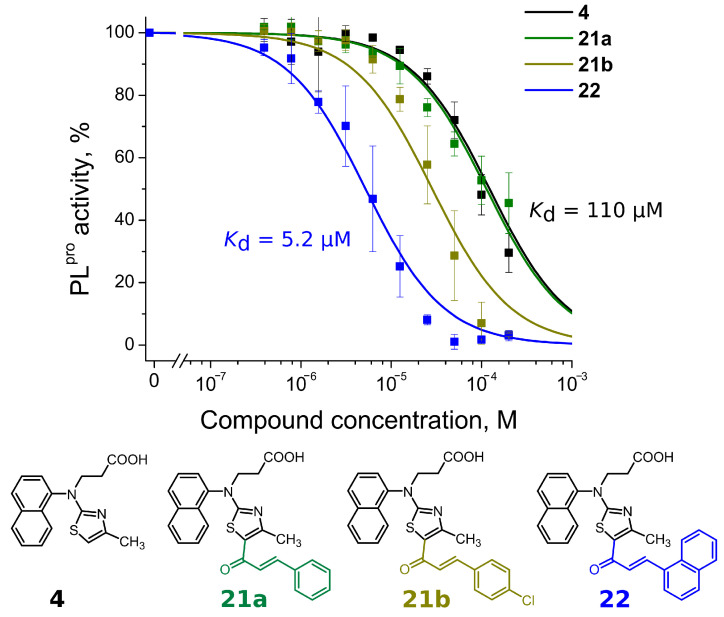
Concentration–response plots for PL^pro^ inhibition with compounds **4**, **21a**,**b** and **22**. Addition of (naphtalen-1-yl)acryloyl fragment at the 5-position (compound **22**) resulted in more than 20-fold stronger affinity than compound **4**.

**Figure 6 ijms-24-13491-f006:**
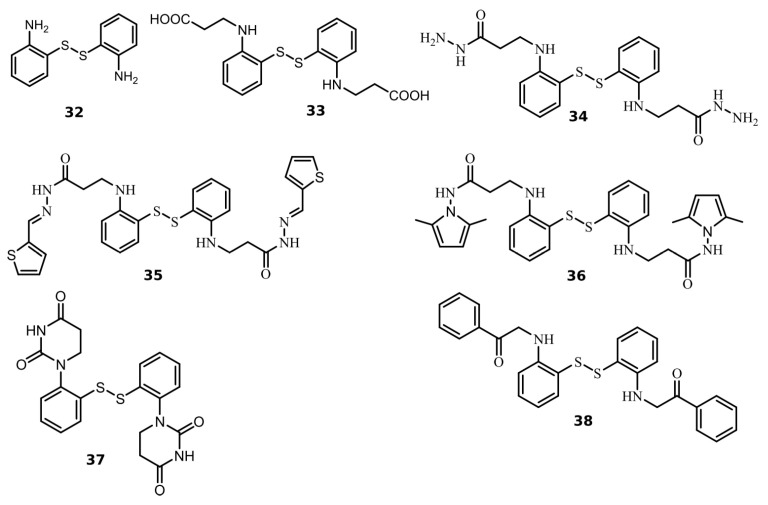
Chemical structures of disulfide compounds discovered as PL^pro^ inhibitors. The synthesis of the compounds and their characterization have previously been described in [32].

**Figure 7 ijms-24-13491-f007:**
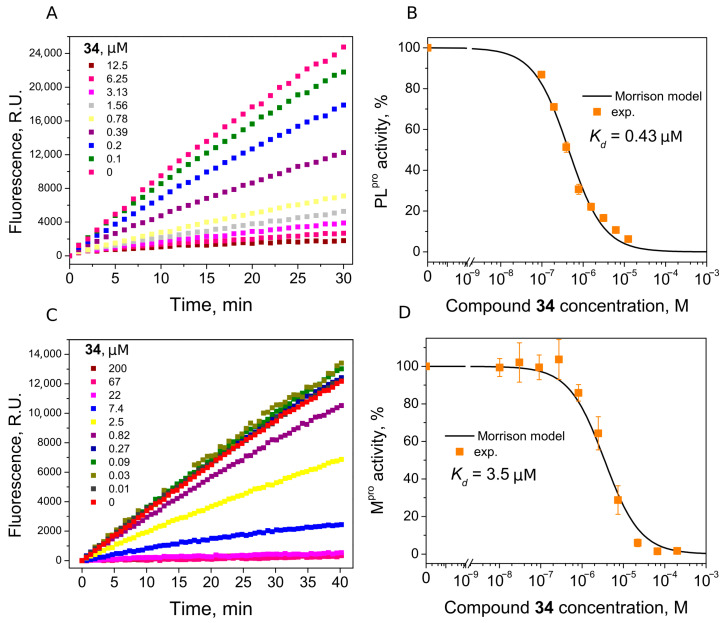
Inhibition of PL^pro^ (**A**,**B**) and M^pro^ (**C**,**D**) activity by compound **34**. (**A**,**C**) Time-dependent product formation in the absence and presence of inhibitor. (**B**,**D**) Dose–response curve of PL^pro^ and M^pro^ inhibition with **34** obtained after fitting curves in panels (**A**,**C**).

**Figure 8 ijms-24-13491-f008:**
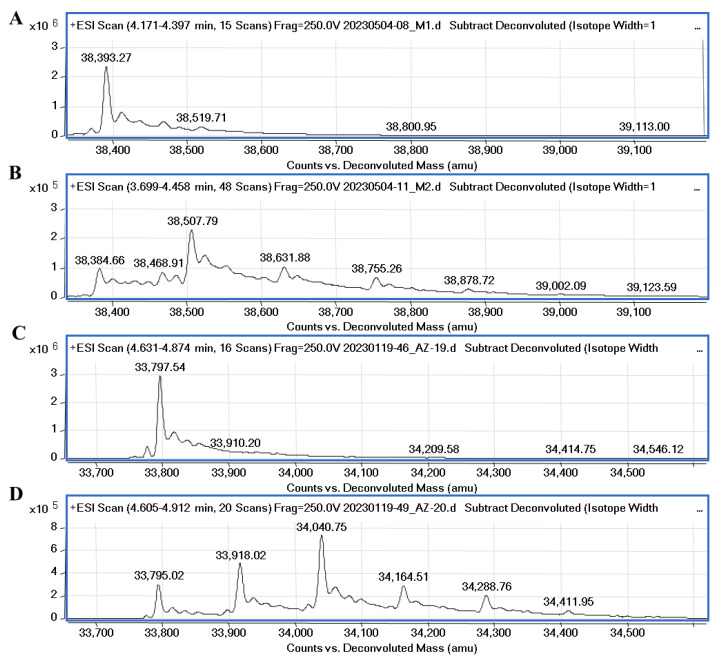
Mass spectra of native PL^pro^, M^pro^ and protease-compound complexes. (**A**)—MS spectra of native PL^pro^ corresponds to the molecular mass of the protein equal to 38,393.27. (**B**)—MS spectra of PL^pro^ incubated with compound **32** (protein/ligand molar ratio 1:5). The difference compared to the molecular weight of PL^pro^ is equal to 123.13 Da. (**C**)—MS spectra of native M^pro^ corresponds to the molecular mass of the protein equal to 33,797.54. (**D**)—spectra of M^pro^ incubated with **32** (protein/ligand molar ratio 1:10). The difference in molecular weight of M^pro^-**32** complex (major peak) compared to the molecular weight of native M^pro^ is equal to 245.73 Da.

**Figure 9 ijms-24-13491-f009:**
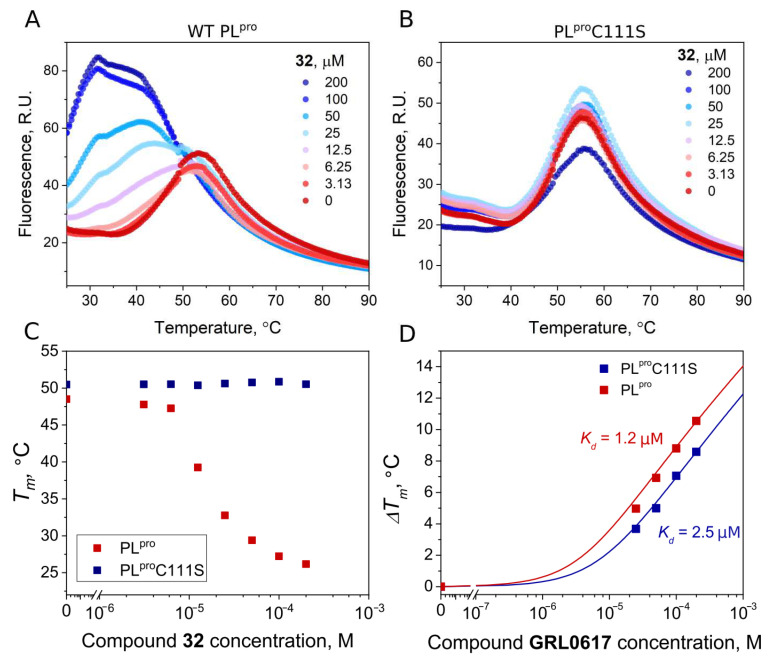
Compound binding measured by FTSA. Raw data of **32** binding to wild-type PL^pro^ (**A**) and mutant PL^pro^C111S (**B**). The thermal destabilization effect (*T_m_* decrease) induced by **32** is seen only for wild-type PL^pro^ (**C**), indicating that the destabilization is attributed to the direct binding of the ligand to the Cys111. (**D**) Dependence of *T_m_* of wild-type and mutant PL^pro^ on concentrations of the added GRL0617 (red squares—wt PL^pro^, blue squares—PL^pro^C111S). Dosing curves were fitted according to [35].

**Table 1 ijms-24-13491-t001:** The dissociation constants *K*_d_ (µM) for compound interaction with PL^pro^ and M^pro^ were determined by enzymatic inhibition assay at 25 °C and pH 7.5.

Compound			*K*_d_, µM	
			PL^pro^	M^pro^
		**1**	68	>200
		**2**	29	>200
		**3**	14	25
**R_1_ group**		**4**	110	>200
		**5**	>200	>200
		**6a**	≥50	>200
		**6b**	≥50	>200
		**6c**	≥50	>200
		**6d**	25	26
		**6e**	20	>200
		**7**	≥50	>200
		**8**	35	>200
		**9**	≥50	>200
		**10**	35	>200
		**11**	>200	>200
		**12**	29	>200

**R_2_ group**	**Short tail**	**13a**	24	6.5
		**13b**	≥50	>200
		**13c**	8.6	8.6
		**14**	9.2	>200
		**15**	≥50	>200
		**16**	≥50	>200
		**17**	>200	>200
		**18**	≥50	>200
		**19**	≥50	>200
		**20**	29	>200

	**Long tail**	**21a**	110	>200
		**21b**	29	17
		**21c**	52	>200
		**22**	5.2	10
		**23**	75	>200
		**24**	8.0	>200

**Two tails/** **scaffold variation group**		**25a**	83	>200
		**25b**	66	>200
		**25c**	58	>200
		**26**	16	>200
		**27a**	35	>200
		**27b**	≥50	>200
		**27c**	15	>200
		**28**	8.8	≥50
		**29**	>200	>200
		**30a**	≥50	>200
		**30b**	≥50	>200
		**30c**	32	>200
		**30d**	≥50	>200
		**31**	≥50	≥50
		GRL0617	1.6	>200

Uncertainty of measurement is approximately 1.8-fold of the *K*_d_.

**Table 2 ijms-24-13491-t002:** The dissociation constants *K*_d_ (µM) for disulfide compounds’ interaction with PL^pro^ and M^pro^ were determined using the enzymatic inhibition assay at 25 °C and pH 7.5.

Compound	*K*_d_, µM	
	PL^pro^	M^pro^
**32**	0.54	2.8
**33**	2.0	>200
**34**	0.43	3.5
**35**	2.5	>200
**36**	1.9	>200
**37**	0.63	>200
**38**	1.9	>200

## Data Availability

Not applicable.

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
