# Peer review of "Screening, Synthesis and Biochemical Characterization of SARS-CoV-2 Protease Inhibitors"

_ijms, 2023, doi:10.3390/ijms241713491_

Round 1
Reviewer 1 Report
The authors of the manuscript titled “Library Screening, Synthesis and Affinity Measurements of SARS-CoV-2 Protease Inhibitors” screened around 1300 compounds using a fluorescence thermal shift assay (FTSA) and identified 53 hits. The identified compounds belong to thiazole and disulfide derivatives, Seven of the disulfide derivatives were identified as efficient PLpro inhibitors.
My overall comment is accepting the manuscript with minor corrections.
1- The titles.
Authors need to re-write the title, for example, “Library screening” is not the correct expression, use a biological screening or screening. Also “ Affinity measurements “ is not a correct expression, use biological investigations.
2- Introduction
2.1. It is a good introduction, but the authors need to add references in some positions, for example, paragraph 31-36 need a reference.
2.2. Line 43 (Numerous drug discovery programs have emerged to find), what do authors mean by drug discovery programs?
3. Results
3.1. most of the figures required adjusted and cantered
3.2. Figure 4B, X-axis is “[13c], M” what is [13C]? , X-axis is the concentration or Kd while 13C is a compound. Authors can write the name of the compound under the graph in the legend while writing the correct label for the X-axis, authors need to do that for the whole manuscript.
3.3. In section 3.2 “Disulfides PLpro and Mpro inhibitors”, I suggest re-write as Disulfide compounds or derivatives and applying this throughout the whole manuscript.
3.4. The font size of the compounds needs to be reduced.
3.5. Figure 5, Authors need to put the compounds in order.
5. Conclusion
The authors write a good brief of the study in the conclusion, I suggest adding the point ( further optimization at 2‘-position on the thiazole ring is required to improve the activity of the compounds as SARS-CoV-2 proteases inhibitors )
Reviewer 2 Report
Bagdonas and colleagues utilize the thermal shift assay to identify compounds from a library of 1300 candidates. The inhibitors, particularly symmetrical disulfides, demonstrated significant inhibitory activity against both PLpro and Mpro, as indicated by their dissociation constants (Kd) in the sub-micromolar range. The findings suggest the possibility of developing these compounds as lead molecules for the development of effective drugs against SARS-CoV-2. Overall, this is a well-conducted study that makes a good contribution to the field of SARS-CoV-2 drug development. I recommend it for publication.
Author Response
We appreciate the reviewer for the positive evaluation of the manuscript.
Reviewer 3 Report
This manuscript discusses the screening and characterization of an in-house library of small molecules, comprising approximately 1,300 compounds. The authors employed biophysical and biochemical approaches to identify the most promising lead molecules against the potential targets of SARS-CoV-2. The hit compounds identified in this study demonstrated inhibition of the two most crucial proteases of the coronavirus. The overall rationale and results are of significant interest to the scientific community, and the manuscript is well-structured. However, before this paper is accepted for publication, the authors should consider the following points:
· The authors have convincingly demonstrated the covalent nature of the interaction between the protein and molecules using mass spectrometry. However, I strongly recommend that the authors also conduct in silico molecular docking studies with the compounds mentioned. It would be beneficial to include a figure showing the optimal poses of the binding site residues that exhibit the lowest binding energy.
· Additionally, though I don't insist, it would be advantageous if the authors could also conduct molecular simulation and dynamics studies to demonstrate how the protease is stabilized or destabilized upon binding with these lead molecules.
The language of the paper also should be thoroughly proofread to remove grammatical errors. Example: In Line 196, protein concentration is written as mg/ml (instead of mg/mL) and also does not mention the incubation temperature.
